# Elevated Levels of IL-33, IL-17 and IL-25 Indicate the Progression from Chronicity to Hepatocellular Carcinoma in Hepatitis C Virus Patients

**DOI:** 10.3390/pathogens11010057

**Published:** 2022-01-03

**Authors:** Momen Askoura, Hisham A. Abbas, Hadeel Al Sadoun, Wesam H. Abdulaal, Amr S. Abu Lila, Khaled Almansour, Farhan Alshammari, El-Sayed Khafagy, Tarek S. Ibrahim, Wael A. H. Hegazy

**Affiliations:** 1Department of Microbiology and Immunology, Faculty of Pharmacy, Zagazig University, Zagazig 44519, Egypt; hishamabbas2008@gmail.com; 2King Fahd Medical Research Center, Department of Medical Laboratory Technology, Faculty of Applied Medical Sciences, King Abdulaziz University, Jeddah 21589, Saudi Arabia; hsadoun@kau.edu.sa; 3Department of Biochemistry, Faculty of Science, Cancer and Mutagenesis Unit, King Fahd Center for Medical Research, King Abdulaziz University, Jeddah 21589, Saudi Arabia; whabdulaal@kau.edu.sa; 4Department of Pharmaceutics, College of Pharmacy, University of Hail, Hail 81442, Saudi Arabia; a.abulila@uoh.edu.sa (A.S.A.L.); kh.almansour@uoh.edu.sa (K.A.); frh.alshammari@uoh.edu.sa (F.A.); 5Department of Pharmaceutics and Industrial Pharmacy, Faculty of Pharmacy, Zagazig University, Zagazig 44519, Egypt; 6Department of Pharmaceutics, College of Pharmacy, Prince Sattam Bin Abdulaziz University, Al-Kharj 11942, Saudi Arabia; e.khafagy@psau.edu.sa; 7Department of Pharmaceutics and Industrial Pharmacy, Faculty of Pharmacy, Suez Canal University, Ismailia 41552, Egypt; 8Department of Pharmaceutical Chemistry, Faculty of Pharmacy, King Abdulaziz University, Jeddah 21589, Saudi Arabia; tmabrahem@kau.edu.sa

**Keywords:** Hepatitis C virus (HCV), chronic hepatitis C (CHC), hepatocellular carcinoma (HCC), interleukins, IL-33, IL-17, IL-25

## Abstract

Hepatitis C virus (HCV) is one of the most epidemic viral infections in the world. Three-quarters of individuals infected with HCV become chronic. As a consequence of persistent inflammation, a considerable percentage of chronic patients progress to liver fibrosis, cirrhosis, and finally hepatocellular carcinoma. Cytokines, which are particularly produced from T-helper cells, play a crucial role in immune protection against HCV and the progression of the disease as well. In this study, the role of interleukins IL-33, IL-17, and IL-25 in HCV patients and progression of disease from chronicity to hepatocellular carcinoma will be characterized in order to use them as biomarkers of disease progression. The serum levels of the tested interleukins were measured in patients suffering from chronic hepatitis C (CHC), hepatocellular carcinoma (HCC), and healthy controls (C), and their levels were correlated to the degree of liver fibrosis, liver fibrosis markers and viral load. In contrast to the IL-25 serum level, which increased in patients suffering from HCC only, the serum levels of both IL-33 and IL-17 increased significantly in those patients suffering from CHC and HCC. In addition, IL-33 serum level was found to increase by liver fibrosis progression and viral load, in contrast to both IL-17 and IL-25. Current results indicate a significant role of IL-33 in liver inflammation and fibrosis progress in CHC, whereas IL-17 and IL-25 may be used as biomarkers for the development of hepatocellular carcinoma.

## 1. Introduction

The positive single-stranded RNA Hepatitis C virus (HCV) is a member of the family *Flaviviridae* [1]. HCV infection is epidemic, and 170 million people are infected with HCV worldwide. About 60–80% of infected people are chronic patients (CHC) because of the failure of immune system to eradicate the virus. Unfortunately, it has been shown that persistent hepatic inflammation in CHC could progress to liver fibrosis, cirrhosis and eventually to hepatocellular carcinoma in many infected patients [2]. Chronic inflammation is basically due to interaction between the virus and host hepatocytes, leading to both hepatocyte injury and immune system suppression [3,4]. The immune response to HCV is regulated by the T-helper (TH) cells that activate both humoral and cellular responses via the secretion of cytokines, which, in turn, regulate both Th1 and Th2 cells [5,6].

Cytokines are small soluble proteins that are important for intercellular communication among immune cells via binding to specific cell receptors. Moreover, cytokines play a crucial dual role in viral eradication and/or tissue injury during the course of HCV infection [7]. Interleukins (ILs) are a subset of cytokines that are expressed by the white blood cells (leukocytes). More than 50 ILs are encoded by human genomes that are divided into four major groups according to structural features [8]. Interleukine-33 (IL-33), a member of IL-1 family with a structural similarity to IL-18 [9], acts through a receptor which is structurally analogous to other IL-1 receptors [10]. IL-33 has a role in mast cells activation, Th2-cells differentiation [9], and it enhances dendritic cell development in bone marrow cultures [11]. The pro-inflammatory activities of IL-33 have been declared in several inflammation models such as inflammatory bowel disease [12], allergic contact dermatitis [13], allergic asthma [14] and autoimmune diseases such as rheumatoid arthritis and systematic lupus erythematosus [15]. In addition, IL-33 plays an important role in host response to various viral infections including human immunodeficiency virus (HIV), dengue virus (DENV) and HCV [16,17,18]. Importantly, increased IL-33 levels are related to liver damage in CHC patients and to the development of HCV/HBV-induced liver fibrosis [18,19].

The family of IL-17 cytokines is comprised of six members; IL-17A (usually referred to as IL-17), IL-17B, IL-17C, IL-17D, IL-17E (commonly known as IL-25), and IL-17F [20,21]. In addition to T helper 17 (Th17) cells, there are other kinds of innate immune cells that produce IL-17, namely γξT and αβT cells [20]. Th17 cells have been shown to be distinct from classical Th1 and Th2 cells [22]. Functionally, IL-17A (IL-17) plays a protective role in the clearance of infections caused by extracellular bacteria [23,24,25], intracellular bacteria [26,27,28], and fungi [29,30]. In addition, IL-17 is involved in chronic inflammation [22], autoimmunity [31], and the promotion of tumor growth [32]. IL-25 (IL-17E) that has only a 16% sequence similarity with IL-17A plays a distinct role in regulating Th2 response against helminthic parasites and in allergic inflammation [33,34]. Importantly, Th17 cells and their immune mediators such as IL-17 are involved in adaptive immune response developed in chronic hepatitis C. Moreover, they could be associated with liver damage, degree of hepatic inflammation as well as disease progression [35,36,37]. Based on the above, the current study hypothesized that IL-17, particularly IL-17A and IL-25, besides IL-33 could play a role in the progression of HCV chronicity and the development of hepatocellular carcinoma as a consequence.

In the present study, the serum levels of IL-33, IL-17 and IL-25 were determined in patients with chronic hepatitis C (CHC) and hepatocellular carcinoma (HCC) and compared to healthy controls (C). In addition, the correlation between serum levels of IL-33, IL-17 and IL-25 and viral load, degree of hepatic inflammation and disease progression was characterized. Furthermore, the levels of serum fibrosis markers such as gamma-glutamyl transferase (γ-GT) and bilirubin were evaluated and correlated to the levels of tested interleukins. Transforming growth factor-β (TGF-β) regulates the plasticity of macrophages during liver fibrosis and plays a crucial role in liver fibrosis development [38,39]. On the other hand, IL-10 plays a dual role in immune suppression or stimulation and is associated with the prognosis of CHC to HCC as reviewed in [40,41]. In order to expect the possible mechanism of IL-17, IL-25, and IL-33 in prognosis of HCV, the levels of TGF-β and IL-10 were evaluated and correlated to the levels of tested interleukins. Moreover, zinc (Zn) deficiency has been observed in chronic hepatitis C patients [42,43], and it was shown to inhibit the replication of viral hepatitis A [44]. Interestingly, Zn improved the outcome of chronic HCV and enhanced the liver viability [43,45]. Furthermore, Zn modulates the expression of some involved interleukins such as IL-17 in chronic HCV patients [46]. Therefore, Zn levels were evaluated as an additional indicator for liver viability. The potential role of these interleukins in HCV pathogenesis and the possibility of their use as biomarkers for disease progression has been evaluated as well.

## 2. Results

### 2.1. Serological Diagnosis

A total of 146 patients with CHC and 45 patients diagnosed with HCC in addition to 60 healthy controls were recruited in the current study. In addition to the pathological examination of liver tissues, the liver enzymes ALT and AST were used to evaluate liver functionality in CHC and HCC patients and compared with healthy controls. The serum concentrations of the liver enzymes AST and ALT were significantly higher in CHC and HCC patients than controls. Furthermore, in contrast to healthy controls, anti-HCV antibodies were determined in both CHC and HCC patients. High levels of HCV RNA were observed in both CHC and HCC patients (Table 1).

### 2.2. Interleukin Levels in CHC and HCC

The serum levels of IL-33, Il-17 and IL-25 were detected by ELISA in patients and healthy individuals. IL-33 serum levels were significantly elevated in patients with CHC and HCC compared to healthy controls (*p* < 0.0001). However there was no significant difference in serum levels of IL-33 between CHC and HCC patients (Figure 1A). Similarly, serum levels of lL-17 were significantly higher in CHC and HCC than controls (*p* < 0.0001). In addition, IL-17 level was significantly higher (*p* < 0.0001) in HCC than those in CHC patients (Figure 1B). Finally, serum IL-25 levels were significantly higher in HCC than those in both CHC patients and healthy controls (*p* < 0.0001). However, there was no significant difference in the serum levels of IL-25 between CHC patients and controls (Figure 1C). Furthermore, the levels of the tested interleukins were assayed in liver biopsies. In compliance with the interleukin levels in serum, there was no significance difference between IL-33 levels in CHC and HCC patients. Meanwhile, the levels of IL-17 and IL-25 increased significantly in HCC as compared to CHC patients (Figure 2).

### 2.3. Correlation between Interleukins Levels and Liver Health Condition

The correlation between serum levels of tested interleukins and liver enzymes was determined in CHC and HCC patients. The IL-33, IL-17 and IL-25 serum levels significantly correlated to both aspartate transaminase (*AST*) and alanine transaminase (ALT) enzymes. The serum levels of IL-33 correlated more significantly to ALT enzyme concentrations (Pearson r = 0.85; *p* < 0.0001) than both IL-17 and IL-25 (Pearson r = 0.18; *p* = 0.0125 and Pearson r = 0.23; *p* = 0.0015, respectively) (Figure 3A–C). Similarly, the IL-33 serum levels correlated more significantly to AST enzyme concentrations than IL-17 and IL-25 (Pearson r = 0.76; *p* < 0.0001, Pearson r = 0.29; *p* < 0.0001 and Pearson r = 0.34; *p* < 0.0001; respectively). For assessment of the prognosis to liver fibrosis, further liver fibrosis serum markers were assayed and their correlations to tested interleukins were evaluated. Gamma-glutamyl transferase (γ-GT) and bilirubin serum levels besides liver enzymes are among the most used liver fibrosis markers, as reviewed in [47]. Interestingly, the serum IL-33 level correlated significantly to the levels of both bilirubin and γ-GT (Pearson r = 0.89; *p* < 0.0001 and Pearson r = 0.86; *p* < 0.0001, respectively). On the other hand, the level of γ-GT did not correlate to the levels of IL-17 and IL-25 (Pearson r = −0.03; *p* = 0.6707 and Pearson r = 0.10; *p* = 0.1346, respectively). The level of bilirubin correlated significantly to the IL-25 (Pearson r = 0.14; *p* = 0.0454), while it did not corelated to IL-17 level (Pearson r = 0.10; *p* = 0.1369) (Figure 3D,E). Transforming growth factor-β (TGF-β) is a powerful stimulus for collagen formation and its role in promoting liver fibrosis is well documented [39,48]. Interestingly, the serum level of IL-33 correlated significantly to the serum level of TGF-β1, in contrast, there were no significant correlations between levels of IL-17 or IL-25 and TGF-β1 (Pearson r = 0.59; *p* < 0.0001, Pearson r = −0.04; *p* = 0.5459 and Pearson r = 0.02; *p* = 0.7744; respectively) (Figure 3F). For further investigation of the role of the tested interleukins in prognosis of liver inflammation and fibrosis, the levels of IL-10 was assayed and correlated to the level of IL-33, IL-17, and Il-25. It was shown that the levels of IL-10 correlated significantly to the level of IL-33, Il-17 and Il-25 (Pearson r = 0.78; *p* < 0.0001, Pearson r = 0.25; *p* < 0.0001, Pearson r = 0.27; *p* < 0.0001; respectively) (Figure 3G).

Moreover, the Metavir score was employed to evaluate the fibrosis stage of liver tissues in CHC and HCC patients. Twenty-one patients showed no fibrosis (F0), 41 patients showed portal fibrosis without septa (F1), 78 patients showed portal fibrosis with a few septa (F2) and, finally, 51 patients showed numerous septa without cirrhosis (F3). As shown in Figure 4, the IL-33 serum level was significantly higher in highly fibrotic liver patients (F2/F3) than in those with minimal fibrosis signs (F0/F1) (*p* < 0.0001). On the other hand, IL-17 and IL-25 serum levels were not significantly influenced by the fibrosis stage of liver tissues (*p* = 0.20 and 0.05; respectively).

In addition, the serum zinc level as a parameter for the liver viability was measured using the colorimetric method. Zinc levels ranged from 70 to 150 µg/dl in healthy controls. However, serum zinc levels significantly dropped in CHC patients in comparison to healthy controls (*p* < 0.0001). In addition, a significant decrease in serum zinc level was observed in HCC patients in comparison with CHC patients (*p* < 0.0001) (Figure 5A). Additionally, there was a correlation between serum zinc level and liver fibrosis. Serum zinc level significantly decreased in more fibrosis scores (*p* = 0.004) (Figure 5B). Similarly, there was a negative correlation between serum levels of tested interleukins; IL-33, IL-17 and IL-25 and serum zinc levels (Pearson r = −0.5547 *p* < 0.0001, Pearson r = −0.2527 *p* < 0.0001 and Pearson r = −0.3186 *p* < 0.0001; respectively) (Figure 5C).

### 2.4. Correlation between Interleukins Levels and Viral Load

The serum HCV RNA load was detected by quantitative PCR assay both in CHC and HCC patients, and correlated to the serum levels of tested interleukins; IL-33, IL-17 and IL-25 (Figure 6). Serum IL-33 level correlated significantly to serum HCV RNA level (Pearson r = 0.84; *p* < 0.0001). However, no correlation was observed between serum levels of IL-17 or IL-25 and viral load (Pearson r = 0.11; *p* = 0.11 and Pearson r = 0.14; *p* = 0.05, respectively).

## 3. Discussion

The immune responses to HCV infection are complicated, and mechanisms associated with virus clearance and disease resolution such as chronicity, development of hepatocellular carcinoma and liver damage are still not fully understood. Periodical monitoring of liver enzymes such as ALT and AST, in addition to the determining of viral loads, are performed as a routine protocol for detection of anti-HCV positive individuals. Furthermore, pathological analysis of liver and screening for tumor biomarkers are required for patients with elevated viral loads and/or clinically suspicious activity of the virus [36,49]. In this context, the current study aimed to correlate cytokine levels and disease progression from chronicity to carcinoma in HCV patients. The levels of pro-inflammatory interleukins IL-33, IL-17 and IL-25 were determined in serum and liver tissues of both CHC and HCC and compared with their levels in healthy controls.

The tested interleukins IL-17, IL-25, and IL-33 were found to be significantly correlated to the liver enzymes ALT and AST, taking into consideration that IL-33 in particular was more correlated to an increase in liver enzymes. IL-33 is a member of the IL-1 family of cytokines that can activate both nuclear factor-κB (NF-κB) and mitogen-activated protein (MAP) kinase signaling pathways. This could result in promoting Th2 response and evolving related cytokines through the attaching to the receptor complex composed of IL-1 RAcP (IL-1 receptor accessory protein) and ST2 receptor (expressed on Th2 and mast cells) [9,50,51]. Furthermore, IL-33 has been shown to be an important factor in the pathogenesis of some viruses including HIV [17] and dengue virus [16]. Similarly, IL-33 contributes significantly in the pathogenic course of acute hepatitis activated by the plant lectin concanavalin A (ConA) [52].

In agreement with previous studies, current findings show that IL-33 levels increased significantly in patients with elevated viral loads and liver fibrosis. For instance, Wang et al., have indicated that higher levels of IL-33 are related to the liver damage in CHC patients [18]. Moreover, the IL-33 over expression is correlated to the development of HCV/HBV-induced liver fibrosis [19,53]. More importantly, current results showed that an increase in IL-33 levels did not change by progress of the disease from chronicity to hepatic cancer; however, it increased significantly by liver fibrosis. The correlation between serum viral loads and IL-33 is not clear in different viral infections. For instance, at higher viral loads in HCV and Dengue fever, elevated levels of IL-33 were clearly detected [16,18]. In contrast, the increase in HIV viral load was not associated with a mutual increase in IL-33 levels [17]. There is no established mechanism that clearly describes how IL-33 affects the viral load or induces fibrosis in liver tissues [54]. It was suggested that IL-33 is a modulator of immune inflammation/activation in liver pathogenesis that could be used to discriminate the different stages of HCV. Wang et al., reported that elevation of IL-33 levels is positively associated with the development and progression of liver fibrosis [18]. Imbalances in the Th1/Th2 response were first observed in the hepatic pathogenesis, and it was shown that IL-33/ST2 is upregulated in hepatic failure. However, it was suggested that ST2 might be employed as a Th1 cell activation marker that may explain the IL-33/ST2 role in progression of liver fibrosis [54]. Marvie et al., established an association between IL-33 and ST2 overexpression and the development of liver fibrosis in CHC [19]. It can be suggested that elevated IFN-γ and IL-6 serum levels in CHC patients might indicate the continual viral replication and pathogenic progression [18]. IL-33 has been shown to promote IFN-γ production by NK cells. Theses proinflammatory cytokines IFN-γ and IL-6 play important roles in both clearance of infected HCV and in liver injury [55].

IL-17 and IL-25 (refers to IL-17A and IL-17E, respectively) are members of the IL-17 family, which are produced primarily by Th17 cells (reviewed in [20]). IL-17 has a role in the evolved immune response in bacterial, fungal and viral infections, autoimmunity, chronic inflammation as well as tumor growth [22,23,24,25,26,27,28,29,30,31,32,56]. On the other hand, IL-25 is involved in allergy, asthma and parasitic infections [33,34,57]. In spite of low sequence similarity between IL-17 and IL-25, both exhibit pro-inflammatory roles in different immunological aspects. This could suggest the use of both IL-17 and IL-25 as biomarkers and therapeutic targets [20,58,59,60].

Contrarily to IL-33, current findings indicate that serum levels of IL-17 and IL-25 did not increase in HCV patients with higher viral loads or liver fibrosis. These results are in accordance with previous studies. Cabral et al., have shown that neither IL-17 nor IL-25 were produced significantly by peripheral blood mononuclear cells stimulated by HCV antigens and did not correlate to HCV viremia [61]. On the other hand, IL-17 levels in T0 cells, stimulated with core antigen, were found to be directly correlated with viral loads and increased in patients with advanced fibrosis [61]. Furthermore, Cachem et al., have shown that different core-specific T-cells were produced in CHC especially in those patients with advanced liver damage. In this context, the authors indicated that IL-17 levels (as a part of other secreted ILs as IL-6, IL-10, IL-21, IL-1β and INF-γ) increased in CHC and its level correlated directly with the degree of liver fibrosis [62]. Similarly, many previous studies have demonstrated that IL-17 could be involved in liver damage and chronic HCV infection [35,36,37,63,64]. In the current study, IL-17 serum levels increased significantly in patients suffering from CHC and HCC, which indicates its role in HCV pathogenesis. Additionally, a significant increase in IL-17 levels in patients suffering from HCC compared with CHC patients suggests a link between IL-17 levels and liver cancer development. It is worthy of note that the results indicate that there was no correlation between IL-17 serum level and either viral load or liver fibrosis degree. The level of pro-inflammatory IL-25 did not increase by stimulation of peripheral blood mononuclear cells or T0 by different HCV antigens [61]. Furthermore, in comparison with healthy controls, IL-25 serum levels did not correlate with viral load nor with increased in CHC patients. In contrast with current findings, the serum level of IL-25 has been shown to increase in patients suffering from hepatic cancer (HCC) as compared with CHC. Li et al., showed that IL-25 serum levels was significantly raised in HCC patients; however, no direct correlation between IL-25 and the development of HCC cells has been observed [58]. Tumor-associated macrophages (TAM) and their related cytokines are closely linked with progression of HCC [65,66]. Type 2 macrophages (M2) are the main TAMs that contribute to hepatocellular carcinoma growth and metastasis [66,67]. Thus, it is expected that increased IL-25 level in HCC could be due to the positive correlation between IL-25 level and M2. Moreover, IL-25 activates macrophages promoting HCC cell migration and invasion [58].

Several parameters can be adopted to assess the liver cells viability, ranging from clinical tools such as liver imagining to different laboratory parameters such as albumin, bilirubin, prothrombin time, and γ-glutamyl transferase (γ-GT) [68,69]. Bilirubin, in addition to the liver enzymes ALT and AST, is routinely employed to assess the liver viability, and high levels indicate liver diseases and are used as primary liver fibrosis markers [47]. Moreover, Bilirubin was shown to be an index for elevated liver fibrosis among HBV carriers [70]. In the current study, the levels of liver enzymes and bilirubin were significantly correlated with the level of IL-33. The liver contains the highest levels of γ-GT, and its elevated serum level indicates liver damage. The elevated level of γ-GT as an indicator for prognosis of viral hepatitis and its correlation to liver fibrosis was documented [71,72]. Current findings showed a significant correlation between IL-33 levels and γ-GT levels, in contrast to IL-17 or IL-25 that did not correlate with γ-GT levels. These findings are in compliance with the histopathological results which showed high IL-33 levels in fibrotic livers. Furthermore, these results agree with an independent study that showed the association between IL-33 overexpression with γ-GT in biliary atresia [73].

Additionally, the pleiotropic cytokine IL-10 level was assayed and correlated to the levels of tested interleukins. The dual roles of IL-10 are known in the immune system, including both suppression and stimulation [41]. In a meta-analysis it was shown that serum IL-10 is associated with a worse prognosis in HCV patients and is significantly increased in HCC and CHC. Moreover, the IL-10 showed an immunosuppressive role on circulating dendritic cells in HCC patients, which may indicate tumor immune evasion [74]. In agreement with these studies, present results showed that IL-10 was significantly correlated to the surge of the three tested interleukins IL-33, IL-17, and IL-25 in both CHC and HCC patients. Zhang et al., showed that IL-33 could promote the production of IL-10 in macrophage-derived foam cells, and proved the correlation between IL-33 and IL-10 [75].

The family transforming growth factors (TGF-β) controls numerous cellular responses which are essential for the homeostasis of most human tissues. For instance, TGF-β was identified as a profibrogenic cytokine because of its role in activation of hepatic stellate cells (HSC), the principal producer of the extracellular matrix during liver fibrosis. There are several supposed mechanisms of activation of HSC by TGF-β [38,76]. In order to understand the association between IL-33 and liver fibrosis, the level of TGF-β1 was measured and correlated to the IL-33 level. A significant correlation was found between levels of IL-33 and TGF-β levels, which may explain its role in liver fibrosis. The IL-33–TGF-β feedforward loop was explained by Taniguchi et al. IL-33 promotes the activation of macrophages that express the high-affinity immunoglobulin E receptor FcεRIα that in turn promote TGF-β to upregulate the expression of IL-33 [77]. On the other hand, our data revealed that there were no significant correlations between IL-17 or IL-25 and TGF-β. Bergis, et al., showed that no significant difference in IL-33 serum levels was found in HCC compared to liver cirrhosis patients [78]. This is in great compliance with our findings that IL-33 is associated with CHC but not furtherly induced in HCC, and it is a negative prognostic factor in HCC.

In addition to the above mentioned interleukins, A link between serum zinc level and liver diseases has been reported [79]. Serum level of zinc has been used to indicate the liver health state, where zinc concentration in serum was found to be lower in patients suffering from CHC. The worsening of liver condition from chronic hepatitis to cirrhosis and hepatocellular carcinoma has been shown to severely affect serum zinc levels [80,81]. These facts not only gave the chance to use zinc serum level clinically as an indicator for liver viability, but also to use zinc in enhancing interferon response in CHC patient therapy [82]. Herein, zinc serum levels were measured both in HCV patients and healthy controls as an indicator for liver viability. The results indicate that serum zinc levels significantly decreased by progress of liver disease from chronicity to hepatocellular carcinoma and markedly decreased by increase of liver fibrosis. Moreover, a negative correlation between IL-33, IL-17 and IL-25 serum levels and zinc concentrations in blood has been found.

As a conclusion, the serum levels of pro-inflammatory cytokines IL-33, IL-17 and IL-25 were measured in patients suffering from CHC or HCC and were compared with healthy controls. The results of the current study could be helpful in evaluating the role of these pro-inflammatory cytokines in HCV pathogenesis and the development of hepatic cancer. A correlation was found between IL-33 and IL-17, on one hand, and the chronicity of HCV and liver viability on the other. Current data show that, in contrast to IL-17, serum levels of IL-33 significantly increased in patients with liver fibrosis. However, IL-17 levels were significantly higher in patients suffering from HCC than those with CHC. There was no correlation between IL-25 levels and viral loads, chronicity or fibrosis degree. However, IL-25 levels significantly increased in patients suffering from HCC, which could be explained by its capability to activate M2 cells; the most important TAMs. Current studies highlight the role of the IL-33, IL-17, and IL-25 in the pathogenesis of HCV and its progression to hepatic carcinoma. Future work is required in order to corroborate these findings and deeply explore the involved mechanisms.

## 4. Materials and Methods

### 4.1. Patients

Blood samples were collected from 146 CHC patients (median age is 43 years old) admitted to El-Ahrar Hospital, Zagazig, Egypt. Individuals tested as positive anti-HCV antibodies with serum containing HCV-RNA, for at least six months, were diagnosed as CHC patients [83]. Another group comprising 45 patients (median age is 51 years old) suffered from CHC, and their liver biopsies were pathologically examined showing hepatocellular carcinoma (HCC) was included in this study. These patients were further confirmed as having hepatocellular carcinoma by both radiological examination and serological determination of tumor markers. As a control (C), a group of 32 healthy individuals in a similar age range (median age is 43 years old) was included in the study. Individuals that were found to be infected with hepatitis B virus (HBV), hepatitis D virus (HDV), or human immunodeficiency virus (HIV), or those suffering from metabolic liver disease or autoimmune hepatitis who received treatment were excluded from the current study. Furthermore, it is worthy to mention that patients who received anti-viral treatments were excluded from this study to exclude any other factors including treatments that could have affected the levels of tested interleukins in selected patients. The demographic and clinical characteristics of selected individuals are presented in Table 1. The collected blood samples from selected individuals and their sera were prepared and stored at −80 °C.

### 4.2. Serological Examination of Collected Sera

Collected sera were examined according to the protocol used in hospital for diagnosis of HCV patients or suspected individuals. Levels of the liver enzymes **(AST and ALT**) and bilirubin were measured using a Biochemistry Automatic Analyzer (Roche Diagnostics, Branchburg, NJ, USA). The concentrations of HCV antibodies were detected using the Enzyme Linked Immunosorbent Assay (ELISA) kit (Abbott Laboratories, Abbott Park, USA). Patients showed positive HCV antibodies were subjected to further clinical examinations and the serum level of HCV RNA was detected. Serum levels of γ-GT, TGF-β, and IL-10 were ELISA assayed according to the instruction of manufacturer’s instructions using MyBioSource, #MBS9346542, #MBS175889 and #MBS8800144, respectively (San Diego, CA, USA). The HCV viral load assayed by a quantitative PCR assay applying a luciferase quantization detection kit (Roche Amplicor, Basel, Switzerland) according to the provided protocol. The detection limit of HCV viral RNA was 300 copies/mL.

### 4.3. Measurement of Zinc Levels

Zinc levels in patients’ sera were detected as an indicator for liver cell viability by colorimetric analysis kit (Biodiagnostic, Cairo, Egypt) according to the provided assay protocol. Zinc was complexed by zincon (2-carboxy-2′-hydroxy-5-sulfoformazyl-benzene) at alkaline pH, and the formed colored complex was spectrophotometrically assayed at a wavelength of 610 nm. A standard curve was established by using the provided standard zinc concentrations. Samples were measured in-triplicate and the serum zinc concentrations were evaluated in correlation to the established standard.

### 4.4. Histopathological Examination of the CHC Patients’ Livers

The pathological examinations of diagnosed HCV patients were kindly provided by the Department of Pathology, El-Ahrar Hospital, Zagzig, Egypt. The biopsied liver tissues were processed according to the typical histology protocols, where tissues were initially fixed using formalin (10%), inserted in paraffin, sectioned at thickness 5 μm by and rotatory microtome, and finally stained with hematoxylin-eosin and Van Gieson. The necroinflammation and fibrosis in tested tissues were analyzed according to the Knodell scoring systems and modified by Ishak using Histologic Activity Index (HAI). The Metavir score is designated particularly for CHC patients; the activity levels or amounts of inflammatory infiltrates are scored from 0 to 3, and extent of fibrosis is scored from 0 to 4. The degree of liver fibrosis was evaluated using the Metavir scoring system [49]. The progress of chronic liver disease progresses through different stages of fibrosis to cirrhosis. The patients’ livers that showed no fibrosis were considered (F0). Stage 1 (F1) fibrosis was observed as portal fibrosis without septa. Stage 2 fibrosis (F2) in which fibrosis is observed only in periportal or perivenular areas. Stage 3 fibrosis (F3) is also known as “bridging fibrosis” that extends across lobules, between portal areas, and between portal areas and central veins. Cirrhosis or stage 4 fibrosis (F4) is considered when fibrosis progresses to and distorts the liver architecture with the formation of nodules.

### 4.5. Quantification of IL-17, IL-25 and IL-33

The serum concentrations of interleukins; IL-17, IL-25 and IL-33, in both patients and healthy controls were determined according to manufacturer’s instructions using human IL-33 ELISA Kit (Cloud-Clone Corp., #SEB980Hu), human IL-17 ELISA Kit (Cloud-Clone Corp., #SEA063Hu) and human IL-25 ELISA Kit (Cloud-Clone Corp., #SEB694Hu), respectively. The tested sera were diluted 1:2 prior to ELISA assay and the interleukins’ concentrations were determined in accordance to the established standard curves using recombinant IL-17, IL-25 and IL-33. The assay range of the ELISA kits was 15.6–1000 pg/mL according to the provided protocol.

The concentrations of interleukins IL-17, IL-25, and IL-33, in HCC and CHC liver samples were determined according to manufacturer’s instructions using a human IL-33 ELISA Kit (MyBioSource, #MBS733754), human IL-17 ELISA Kit (MyBioSource, #MBS849335) and human IL-25 ELISA Kit (MyBioSource, #MBS4500319), respectively. The biobased liver tissues were rinsed with ice-cold PBS, minced to small pieces, and homogenized in PBS with a glass homogenizer on ice. The obtained suspensions were ultrasonicated and centrifuged for 15 min at 5000 rpm. The used reagents were prepared and the tested samples were diluted 1:2 prior to ELISA assay and the interleukins’ concentrations were determined in accordance with the established standard curves. The assay range of the IL-17 ELISA kit was 31.25–2000 pg/mL, while for the IL-33 and IL-25 ELISA kits they were 15.6–1000 pg/mL according to the provided protocols.

### 4.6. Statistical Analysis

The ELISA determination of tested interleukins was performed in triplicates and the results were presented as median and range unless specified. The significant difference between groups was analyzed by a Student’s t-test and one-way ANOVA test (Graphpad Prism 8 software). A Pearson rank correlation test was employed to assess the relationship between variables. A *p* value < 0.05 was considered statistically significant.

## 5. Conclusions

The levels of cytokines in patients suffering from CHC and/or HCC have been determined in the present study in order to characterize new biomarkers that could be helpful in the diagnosis and prediction of hepatic carcinoma. The results indicate a possibility of the use of IL-33 as a biomarker for liver fibrosis progress in patients suffering from CHC, whereas IL-17 and IL-25 could be used as biomarkers for the development of hepatocellular carcinoma. Such findings might open the way for further studies to explore the exact mechanism underlying the role of tested interleukins in the progression of HCV from chronicity to hepatocellular carcinoma.

## Figures and Tables

**Figure 1 pathogens-11-00057-f001:**
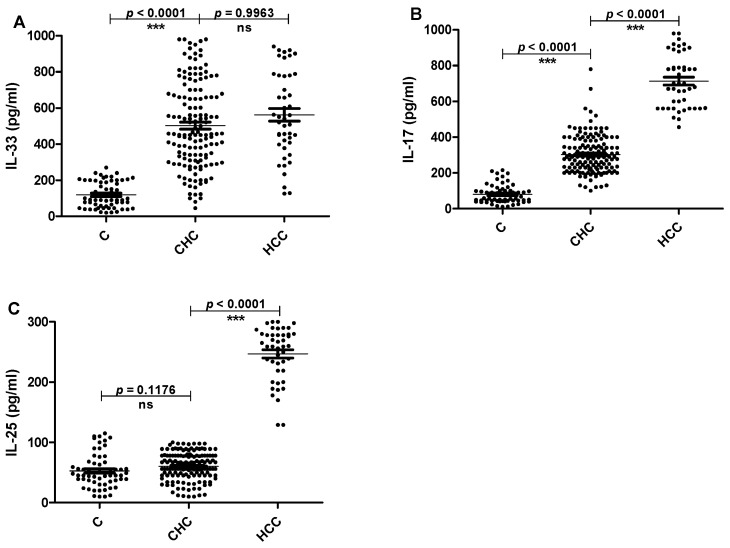
The serum IL-33, IL-17 and IL-25 levels. The horizontal lines refer to the median values of different groups. (**A**)The basal levels of IL-33 in serum of CHC, HCC and C groups. (**B**) The basal levels of IL-17 in serum of CHC, HCC and C groups. (**C**) The basal levels of IL-25 in serum of CHC, HCC and C groups. C: healthy controls, CHC: patients with chronic hepatitis C and HCC: patients with hepatocellular carcinoma.

**Figure 2 pathogens-11-00057-f002:**
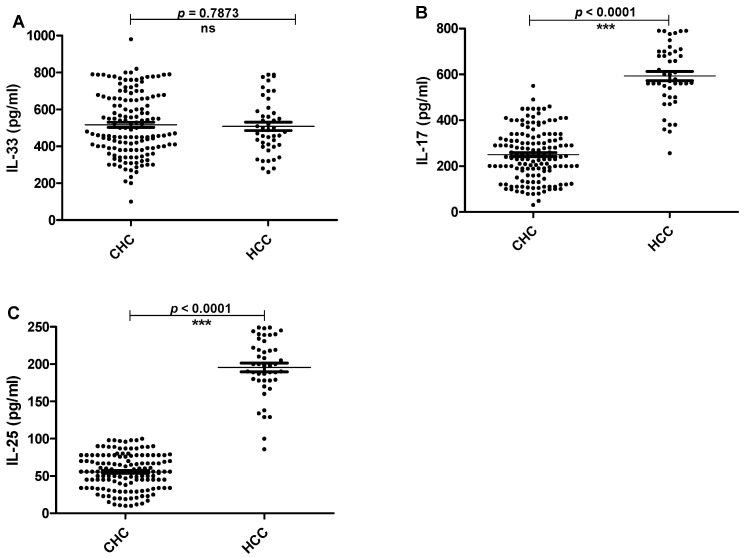
The IL-33, IL-17 and IL-25 levels in liver tissues. The horizontal lines refer to the median values of different groups. (**A**)The basal levels of IL-33 in liver tissue homogenates of CHC and HCC groups. (**B**) The basal levels of IL-17 in liver tissue homogenates of CHC and HCC groups. (**C**) The basal levels of IL-25 in liver tissue homogenates of CHC and HCC groups. CHC: patients with chronic hepatitis C and HCC: patients with hepatocellular carcinoma.

**Figure 3 pathogens-11-00057-f003:**
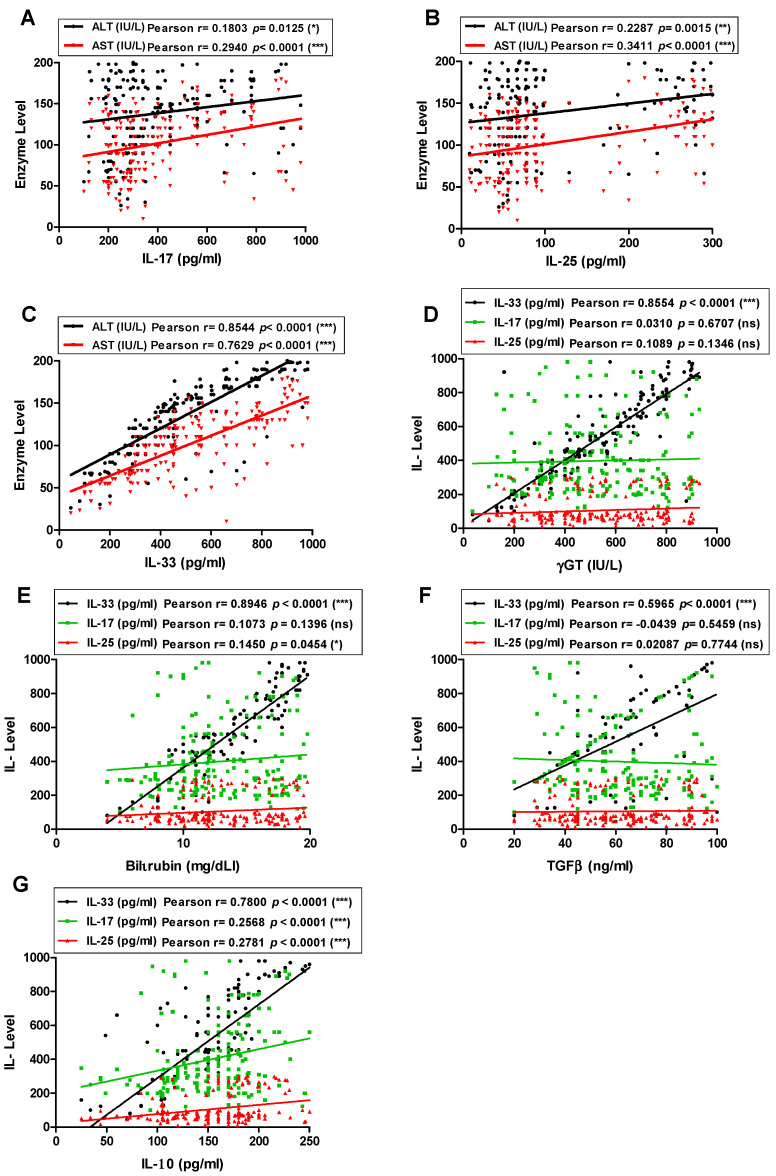
Correlation between the levels of serum interleukins and liver fibrosis serum markers in CHC and HCC patients. (**A**) The correlation between the levels of serum IL-33 and ALT or AST. (**B**) The correlation between the levels of serum IL-17 and ALT or AST. (**C**) The correlation between the levels of serum IL-25 and ALT or AST. (**D**) The correlation between the levels of serum IL-33, IL-17 or IL-25 and Gamma-glutamyl transferase (γ-GT). (**E**) The correlation between the levels of serum IL-33, IL-17 or IL-25 and bilirubin. (**F**) The correlation between the levels of serum IL-33, IL-17 or IL-25 and transforming growth factor-β (TGF-β1). (**G**) The correlation between the levels of serum IL-33, IL-17 or IL-25 and IL-10.

**Figure 4 pathogens-11-00057-f004:**
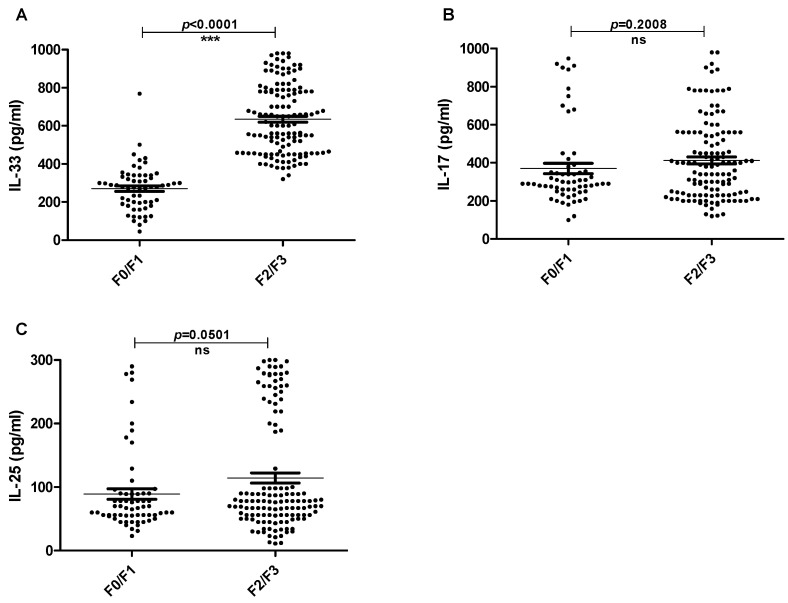
The levels of serum interleukins of CHC and HCC patients with different fibrosis scores. The Metavir scale was used [F0: no fibrosis (n = 21), F1: portal fibrosis without septa (n = 41), F2: portal fibrosis with few septa, (n = 78) and F3: numerous septa without cirrhosis (n = 51)]. Data are presented as the mean values of individual participants from three separate experiments. (**A**) The levels of serum IL-33 of patients with different fibrosis. (**B**)The levels of serum IL-17 of patients with different fibrosis. (**C**) The levels of serum IL-25 of patients with different fibrosis. CHC: patients with chronic hepatitis C and HCC: patients with hepatocellular carcinoma.

**Figure 5 pathogens-11-00057-f005:**
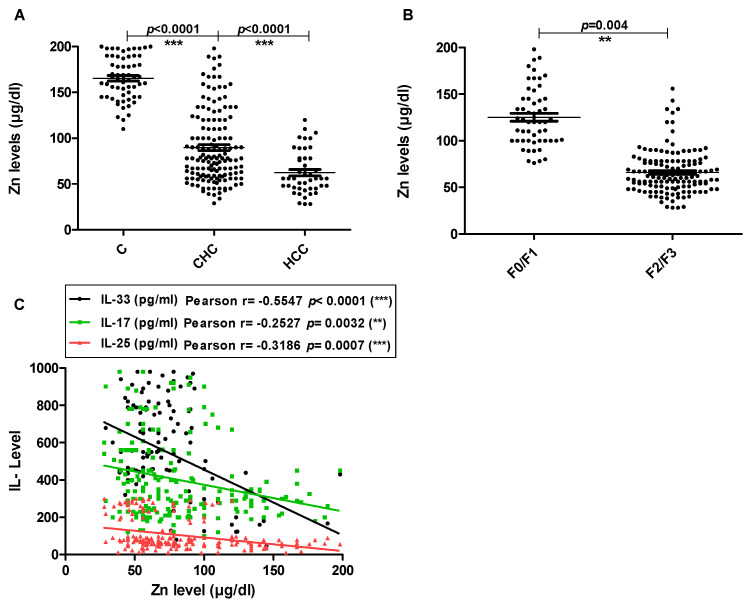
The Zinc levels in the serum. Data are presented as the mean values of individual participants from three separate experiments. The horizontal lines refer to the median values of different groups. (**A**) The basal levels of zinc in serum of CHC, HCC and C groups. (**B**) the levels of serum zinc of CHC and HCC patients with different fibrosis scores. (**C**) The correlation between the levels of serum interleukins and zinc level. C: healthy controls, CHC: patients with chronic hepatitis C and HCC: patients with hepatocellular carcinoma.

**Figure 6 pathogens-11-00057-f006:**
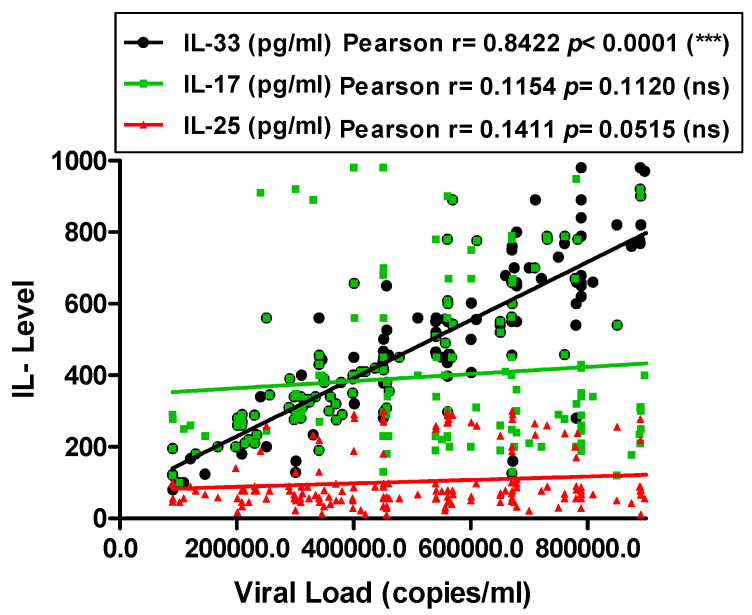
Correlation between serum HCV RNA in CHC and HCC patients and the levels of serum interleukins. Data are presented as the mean values of individual participants from three separate experiments. CHC: patients with Chronic Hepatitis C and HCC: patients with Hepatocellular Carcinoma.

**Table 1 pathogens-11-00057-t001:** Demographic characteristics and clinical features of cases.

Parameter	Chronic Hepatitis C (CHC)	Hepatocellular Carcinoma (HCC)	Control (C)
Number	146	45	60
Age (years)	43 (22–60)	51 (44–65)	41 (30–57)
Sex (M/F)	94/52	31/14	31/29
Viraemia (Log copies/mL)	6.8 (3.11–8)	6.26 (3.14–7.8)	NA *
Anti-HCV	Positive	Positive	Negative
AST (IU/L) **	55.5 (11–123)	81 (11–156)	33.2 (20–65)
ALT (IU/L) **	90.2 (21–301)	110.2 (89–297)	30 (24–48)

* NA: Not Applicable. ** Normal values were considered: ALT = 50 IU/L; AST = 40 IU/L. Data were expressed as median and range.

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
