# Peer review of "Elevated Levels of IL-33, IL-17 and IL-25 Indicate the Progression from Chronicity to Hepatocellular Carcinoma in Hepatitis C Virus Patients"

_pathogens, 2022, doi:10.3390/pathogens11010057_

Round 1

Reviewer 1 Report

In the present study, Askoura et al. investigated the expression of IL-33, IL-17 and IL-25 in patients with chronic HCV infection and HCV-associated HCC, and assessed the correlation of these tested cytokines with clinical index. Overall, the results were preliminary and no mechanism studies were involved. Importantly, chronic HCV infection is now a cruable disease, which was not esstentially correlated with HCC, making the results less important to be published. The highlights of the study was the liver biospy for all patients, however, the authors only assessed Ishak score for the samples.

  1. The IRB approval or waiver statement must be showed in Methods section.
  2.  It is important to examine the tested cytokine levels, especially IL-33 expression, in the liver specimens from CHC and HCC patients.
  3.  The author analyzed IL level and Zn expression. However, no background on this correlation was mentioned on background.

Overall, I did not think the manuscript contained sufficient data to be published in the current version.

Author Response

Dear Reviewer 1,

We are very grateful for your interest in our manuscript and appreciate your valuable and constructive comments and suggestions.

Please find the attached reply to all the points you raised. We hope that you can accept our explanations.

Best Regards,

Authors

Reviewer 2 Report

HCV infection is one of the most epidemic viral infections in the world. About 60-80% of infected people are chronic patients (CHC),and further progress to liver fibrosis, cirrhosis and hepatocellular carcinoma in many of infected patients. In this manuscript, the serum levels of pro-inflammatory cytokines IL-33, IL-17 and IL- 25 were measured in patients suffering from CHC or HCC by ELISA, and results showed that IL-25 level increased in patients suffering from HCC only, and the levels of both IL-33 and IL-17 increased significantly in patients suffering from CHC and HCC. In addition, IL-33 serum level was  associated with the progression of liver fibrosis progression and viral load. The results may help to evaluate the role of these pro-inflammatory cytokines in HCV pathogenesis and development of hepatic cancer. However, since 2014, direct acting antivirals (DAA), as well as a combination of DAA and IFN, have achieved sustained virologic response (SVR) in more than 95% of patients with chronic hepatitis and compensatory cirrhosis, reducing the value of studies that correlate with progression of HCV infection to chronic and liver cancer.  Of course, in a broad sense, chronic liver injury from any cause can progress to fibrosis, cirrhosis, and hepatocellular carcinoma.  The progression of liver disease is regulated by a variety of cytokines, among which cytokines play an important role in liver injury, inflammation and fibrosis, including interferon,Th1 cytokines, Th2 cytokines, Th17 cytokines, IL-1 family and IL-6 family, etc.  Some of these have great therapeutic potential and are currently being targeted in clinical trials for liver disease.  Recent Review (Cell Mol Immunl. 2021;  18(1):18-37.) cited 311 papers on the important role of cytokines in regulating liver injury, inflammation, fibrosis and regeneration, including IL-33, IL-17 and IL-25 involved in this manuscript.

Since a total 191 Blood samples from CHC and HCC patients were collected in this study, this is valuable data.  As far as this study is concerned, the following questions are worth focusing on and supplementing.  

  1. The basis for selecting these three cytokines for detection from numerous cytokines related HCV hepatitis is not sufficient, please supplement it in the preface.
  2. According to the results of biopsy samples, the degree of fibrosis in HCV infected patients was divided into four grades. Are these four stages related to clinical type?    Are there any cases of cirrhosis?    Please add the representative histopathological results.
  3. Are there any differences in viral load between the CHC and HCC ? The results suggested that IL-33 was significantly correlated with viral load.  Please discuss possible mechanisms by which IL-33 may attempt to influence viral load in the discussion. was measured 162
  4. It is too simple using colorimetric method to measure the serum zinc level as parameter for the liver viability, although it is classical.   Suggestion to add other parameter or explain them in the discussion.
  5. In the discussion, the author mentioned the influence of IL-25 on macrophage M2 polarization, and tumor-associated macrophages effect on the progression of liver tumors. If possible, please increase the detection of infiltrating macrophage types in the liver biopsy tissue.6. It can be seen that IL-33 is more clearly correlated with viral load, liver fibrosis and liver enzymes from the manuscript. Please discuss the possible mechanisms in the discussion.  In addition, I do not agreed it is directly related to the activation of TH2 cells and mast cells.

Author Response

Dear Reviewer,

We are very grateful to the reviewer for your valuable and constructive comments, which greatly helped us to improve our manuscript. We have revised the manuscript and provided a point-by-point response to comments. Please find attached the word document.

Reviewer 3 Report

Please specify if you exclude from the control group pleople with comorbidities other than liver disease.

It would be interesting to have some data if patients received antiviral agents and the effect of this treatment.

Author Response

Dear Reviewer 2,

We are very grateful for your interest in our manuscript and appreciate your valuable and constructive comments and suggestions.

Please find the attached reply to all the points you raised. We hope that you can accept our explanations.

Best Regards,

Authors

Round 2

Reviewer 1 Report

No further experiments were performed. The data was still preliminary, which was not sufficient for publication.

Author Response

Dear Reviewer,

We are thankful for your efforts in reviewing this manuscript and respect your opinion. We clearly stated at the end of the Discussion section and Conclusion that our study represents preliminary data and that these results have to be corroborated in the future.

Finally, we are very thankful for the reviewer.

Reviewer 3 Report

If it is possible, please provide some information on HCC stage and how many patients received antiviral therapy.

Author Response

Dear Reviewer,

We are very thankful for the reviewer’s interest in improving our manuscript. We excluded the patients who received any treatment from this study as we found it would be of great significance to exclude any other factors, such as antiviral therapy, that could have affected the levels of the tested interleukins in selected patients. We clearly stated this in Lines 332-335 in the Methods Section in the revised manuscript.

Regarding HCC stage, as the reviewers well know that HCC reveals a considerable geographic and institutional variation throughout the world. Although many staging and/or scoring systems have been proposed, each prognostic system has several benefits and limitations on its own. Therefore, there is currently no globally accepted system for HCC due to the extreme heterogeneity of the disease. Accordingly, we found that the HCC staging may not be highly indicative and instead we depend on the evaluation of the liver fibrosis by the well-known histopathological scoring system.

References:

  • Villanueva A. Hepatocellular Carcinoma. N Engl J Med. 2019 Apr 11;380(15):1450-1462. doi: 10.1056/NEJMra1713263. PMID: 30970190.
  • Tellapuri S, Sutphin PD, Beg MS, Singal AG, Kalva SP. Staging systems of hepatocellular carcinoma: A review. Indian J Gastroenterol. 2018 Nov;37(6):481-491. doi: 10.1007/s12664-018-0915-0. Epub 2018 Dec 29. PMID: 30593649.

Finally, we really thank the reviewer for kind and critical comments.